# A Switched-Element System Based Direction of Arrival (DOA) Estimation Method for Un-Cooperative Wideband Orthogonal Frequency Division Multi Linear Frequency Modulation (OFDM-LFM) Radar Signals

**DOI:** 10.3390/s19010132

**Published:** 2019-01-02

**Authors:** Yifei Liu, Yuan Zhao, Jun Zhu, Jun Wang, Bin Tang

**Affiliations:** 1School of Information and Communication Engineering, University of Electronic Science and Technology of China, Chengdu 611731, China; zy_uestc@outlook.com (Y.Z.); uestczhujun@163.com (J.Z.); bint@uestc.edu.cn (B.T.); 2School of Electronic Science and Engineering, University of Electronic Science and Technology of China, Chengdu 611731, China; Wangjung@uestc.edu.cn

**Keywords:** wideband OFDM-LFM signal, switched-element system, fractional autocorrelation, DOA estimation

## Abstract

This paper proposes a switched-element direction finding (SEDF) system based Direction of Arrival (DOA) estimation method for un-cooperative wideband Orthogonal Frequency Division Multi Linear Frequency Modulation (OFDM-LFM) radar signals. This method is designed to improve the problem that most DOA algorithms occupy numbers of channel and computational resources to handle the direction finding for wideband signals. Then, an iterative spatial parameter estimator is designed through deriving the analytical steering vector of the intercepted OFDM-LFM signal by the SEDF system, which can remarkably mitigate the dispersion effect that is caused by high chirp rate. Finally, the algorithm flow and numerical simulations are given to corroborate the feasibility and validity of our proposed DOA method.

## 1. Introduction

As a novel synthetic aperture radar (SAR) system, the multiple-input multiple-output SAR (MIMO-SAR) utilizes multiple antennas to emit mutually orthogonal waveforms, and employs multiple receiving channels to process the echo signals simultaneously [1,2,3]. Subject to current technical conditions, wideband Orthogonal Frequency Division Multi Linear Frequency Modulation (OFDM-LFM) modulated waveforms are commonly employed in modern MIMO-SAR systems [1,4], which brings challenge to the passive direction of arrival (DOA) estimation techniques.

Passive DOA estimation techniques have been implemented in electronic warfare equipment. In particular, a review of the most commonly used techniques can be found in literatures [5,6,7]. However, most of them are derived for narrowband signals, which cannot handle the wideband signal scenario, i.e., the OFDM-LFM signals. In this paper, we focus on the DOA estimation method for un-cooperative wideband OFDM-LFM radar signals. Overview of existing DOA algorithms [8,9,10,11,12,13] for wideband signals, the common approach is to sample the signals in the frequency domain through the array sensors, then, consider each frequency component into a narrowband signal for processing individually. The broadband beamforming approaches in H. L. Van Trees book [14] utilize arrays with non-uniform element spacing and a time-shift operator to complete decoupling of broadband signals. Although the mentioned methods can function well, they still suffer from huge cost of hardware and computational resources. Therefore, we exploit the switched-element direction finding (SEDF) system to solve the DOA estimation problem for wideband signals without much cost.

The block diagram of the modified SEDF and the target MIMO radar system are drawn in Figure 1. Its primary advantages include reducing the hardware and storage costs, simplifying the channel calibration process and decreasing the computation load [15,16,17]. Moreover, SEDF is also suitable for dealing with long-pulse signals, because there is no need to store the entire pulse in each channel. As shown in Figure 1, we consider a SEDF system with two receiving channels whose name are the reference channel (RC) and the switched channel (SC) respectively. When a signal of interest (SOI) is intercepted, the SC starts to switch in a constant period from antenna #1 to antenna #K. Thus, the signal pulse is split into multiple sub-pulses in the SC. Meanwhile, the data are collected via the RC. In formulating the DOA estimation problem for wideband OFDM-LFM signal on this SEDF system, we found that the steering vector is turned into a discrete time LFM-like vector. Hence, we proposed a modified approach to solve this estimation problem, which is inspired by a recently developed parameter estimation algorithm called Fast Iterative Interpolated Digital Fraction Fourier Transform (FII-DFrFT) [18].

The rest of this paper is organized as follows. In Section 2, we introduce the signal model and the formula derivation for DOA estimation problem. In Section 3, the proposed FII-DFrFT estimator is illustrated in detail. Numerical simulation results are shown in Section 4. Finally, in Section 5 some conclusions are drawn.

## 2. Problem Formulations

Consider an adversary MIMO-SAR with *M* transmitters. This radar employs wideband OFDM-LFM waveforms, which were first introduced into the design of an MIMO radar system by F. Cheng [19]. Afterwards, the signal of the mth transmitter is given as:(1)$$s_{m}\left(t\right)=u_{m}\left(t\right)e^{j2\pi f_{0}t},0\leq m\leq M-1$$
(2)$$u_{m}\left(t\right)=e^{j2\pi \left(mf_{\Delta }t+\frac{1}{2}\gamma_{0}t^{2}\right)}$$
where $f_{0}$ denotes the carrier frequency; $f_{\Delta }$ is the frequency step between two adjacent transmitters; $\gamma_{0}$ stands for the chirp rate. Besides, the bandwidth *B* of the OFDM-LFM signal is defined as $B\overset{\Delta }{\;=}\left(M-1\right)f_{\Delta }+\gamma_{0}T_{p}$, where $T_{p}$ represents the pulse width of $s_{m}\left(t\right)$.

On the contrary, there are K+1 antennas allocated in the SEDF system with interspace $d_{R}$, as shown in Figure 1. Here, we set the intercepted signal via RC as $y_{\mathrm{RC}}\left(t\right)=s\left(t-t_{0}\right)+n_{\mathrm{RC}}\left(t\right)$, where $t_{0}$ represents the propagation time, and $n_{\mathrm{RC}}\left(t\right)$ is the additive Gaussian white noise in RC. Since this paper focuses on the DOA, without loss of generality, it is reasonable to set $t_{0}=0$ for the sake of simplicity of derivations. Meanwhile, to avoid redundancy introductions of other scholars’ existing work, we assume that the estimation for inner pulse parameters and the radio frequency demodulation have already been accomplished by the techniques and algorithms in References [18,20,21,22], while using the collected data in the RC. Moreover, we also assume the incident direction θ and the power of the SOI is stable during the switch period $T_{s}$. Therefore, the OFDM-LFM signal intercepted via the SC can be written as: (3)$$\begin{array}{c} y_{\mathrm{SC}}\left(t\right)=A\sum_{m=0}^{M-1}\sum_{k=1}^{K}s_{m}\left(t-\tau_{k}\right)\mathrm{rect}\left(\frac{t-\left(k-1\right)T_{s}}{T_{s}}\right)+n_{\mathrm{SC}}\left(t\right)\\ =A\sum_{k=1}^{K}\sum_{m=0}^{M-1}\exp\left[j2\pi \left(f_{m}t+\frac{1}{2}\gamma_{0}t^{2}\right)+j2\pi \left(-f_{m}\tau_{k}-\gamma_{0}\tau_{k}t+\frac{1}{2}\gamma_{0}\tau_{k}^{2}\right)\right]\mathrm{rect}\left(\frac{t-\left(k-1\right)T_{s}}{T_{s}}\right)+n_{\mathrm{SC}}\left(t\right)\\ =A\sum_{k=1}^{K}\sum_{m=0}^{M-1}u_{m}\left(t\right)e^{j\varphi_{m}\left(\tau_{k},t\right)}\mathrm{rect}\left(\frac{t-\left(k-1\right)T_{s}}{T_{s}}\right)+n_{\mathrm{SC}}\left(t\right) \end{array}$$
where $\tau_{k}=\left[kd_{R}\sin\left(\theta \right)\right]/c$ is the propagation delay between the #k and #0 antenna, with *c* represents the speed of light; $T_{s}$ is the duration for each switch; $n_{\mathrm{SC}}\left(t\right)$ is the thermal noise in SC; $f_{m}=f_{0}+mf_{\Delta }$; the phase shift $\varphi_{m}\left(\tau_{k},t\right)$ is recast to:(4)$$\varphi_{m}\left(\tau_{k},t\right)=2\pi \left(-f_{m}\tau_{k}-\gamma_{0}\tau_{k}t+\frac{1}{2}\gamma_{0}\tau_{k}^{2}\right)$$
which is time related.

Let us consider a common LFM, whose chirp rate has the quantity of $10^{12}\mathrm{Hz}/s$, while $\tau_{k}$ has the quantity of $10^{-9}$ s. This means that the third term ($\frac{1}{2}\gamma_{0}\tau_{k}^{2}$) in Equation (Equation 4) is almost 0. Thus, we discard this term in the following derivations. Then, ignoring the noise term (its effect will be analyzed in the performance evaluations Section), we can obtain the instantaneous cross correlation between the SC and RC by:(5)$$\begin{array}{c} r\left(t\right)=y_{\mathrm{SC}}\left(t\right)\cdot y_{\mathrm{RC}}^{*}\left(t\right)=A^{2}\sum_{k=1}^{K}\sum_{m=0}^{M-1}u_{m}\left(t\right)\exp\left[j\varphi_{m}\left(\tau_{k},t\right)\right]\sum_{m^{\prime }=0}^{M-1}u_{m^{\prime }}^{*}\left(t\right)\\ =A^{2}\sum_{k=1}^{K}\left[\sum_{m=0}^{M-1}\sum_{m^{\prime }=0}^{M-1}\exp\left[j2\pi \left(m-m^{\prime }\right)f_{\Delta }t+j\varphi_{m}\left(\tau_{k},t\right)\right]\mathrm{rect}\left(\frac{t-\left(k-1\right)T_{s}}{T_{s}}\right)\right] \end{array}$$

The above equation reveals that the interested phase shift terms ($\exp\left[j\varphi_{m}\left(\tau_{k},t\right)\right]$) are mixed with the cross terms ($\exp\left[j2\pi \left(m-m^{\prime }\right)f_{\Delta }t\right]$), which are caused by the multi-component of the intercepted signal. In order to extract the phase shift term, a low-pass filter $h\left(t\right)$ is designed [23] to filter out the cross terms, which ranges from $\pm \exp\left[\pm j2\pi f_{\Delta }t\right]$ to $\exp\left[\pm j2\pi \left(M-1\right)f_{\Delta }t\right]$. Therefore, we can obtain a new baseband signal $x\left(t\right)$ after cross correlation and low-pass filter processing:(6)$$x\left(t\right)=\left\{y_{\mathrm{SC}}\left(t\right)\cdot y_{\mathrm{RC}}^{*}\left(t\right)\right\}⊗h\left(t\right)\approx A^{2}\sum_{m=0}^{M-1}\sum_{k=1}^{K}\exp\left[j\varphi_{m}\left(\tau_{k},t\right)\right]$$

Afterwards, we collect the samples of $x\left(t\right)$ every time when the SC switches the antenna, i.e., at $t=0,T_{s},\cdots ,\left(K-1\right)T_{s}$. Therefore, the sampled data is given by:(7)$$x={\left[x\left(0\right)x\left(T_{s}\right)\cdots x\left(\left(K-1\right)T_{s}\right)\right]}_{K\times 1}^{T}=A^{2}a\left(\theta \right)$$
where the steering vector $a\left(\theta \right)$ is expressed as:(8)$$a\left(\theta \right)={\left[\begin{array}{c} \sum_{m=0}^{M-1}\exp\left[-j2\pi \left(f_{m}\frac{d_{R}\sin\left(\theta \right)}{c}\right)\right]\\ \sum_{m=0}^{M-1}\exp\left[-j2\pi \left(\left(f_{m}+\gamma_{0}T_{s}\right)\frac{2d_{R}\sin\left(\theta \right)}{c}\right)\right]\\ \vdots \\ \sum_{m=0}^{M-1}\exp\left[-j2\pi \left(\left(f_{m}+\left(K-1\right)\gamma_{0}T_{s}\right)\frac{Kd_{R}\sin\left(\theta \right)}{c}\right)\right] \end{array}\right]}_{K\times 1}$$

For the simplicity of derivations, we define $v\overset{\Delta }{\;=}d_{R}\sin\left(\theta \right)/c$. Then, the *k*th entry of x can be further denoted by:(9)$$x\left[k\right]=A^{2}\sum_{m=0}^{M-1}\exp\left[-j2\pi \left(\left(f_{m}-\gamma_{0}T_{s}\right)vk+\gamma_{0}T_{s}vk^{2}\right)\right]$$

It is interesting to find out that comparing with the traditional narrow band representation, the steering vector of OFDM-LFM signal by SEDF system is also a chirp modulated signal, with respect to $k^{2}$. Thus, this spatial signal model brings failure to the regular DOA estimation algorithms such as MUSIC and ESPRIT. Concerning on this, we approach our DOA estimation problem to the parameter estimation for OFDM-LFM signals. Therefore, we define the spatial chirp rate ($\mu_{0}$) and spatial frequency (ω) as $\mu_{0}\overset{\Delta }{\;=}2v\gamma_{0}T_{s}$ and $\omega_{m}=\left(f_{m}-T_{s}\gamma_{0}\right)v$ respectively. Then, Equation (Equation 9) can be simplified as:(10)$$x\left[k\right]=A^{2}\sum_{m=0}^{M-1}\exp\left[-j2\pi \left(\omega_{m}k+\frac{\mu_{0}}{2}k^{2}\right)\right]$$

To solve this estimation problem, we introduce the fast digital algorithm of FrFT [24] as:(11)$$X_{\alpha }\left(\frac{U}{2\Delta x}\right)=\frac{B_{\alpha }}{2\Delta x}e^{j\pi \tan\left(\frac{\alpha }{2}\right){\left(\frac{U}{2\Delta x}\right)}^{2}}\sum_{k=-K}^{K}e^{j\pi \csc\alpha {\left(\frac{U-k}{2\Delta x}\right)}^{2}}e^{j\pi \tan\left(\frac{\alpha }{2}\right){\left(\frac{k}{2\Delta x}\right)}^{2}}x\left(\frac{k}{2\Delta x}\right)$$
where $\Delta x=\sqrt{K}$ and $B_{\alpha }=\sqrt{\left(1-j\cot\alpha \right)}$.

Substituting Equation (Equation 7) into Equation (Equation 11) we can obtain:(12)$$\begin{array}{c} X_{\alpha }\left(\frac{U}{2\Delta x}\right)=\frac{B_{\alpha }}{2\Delta x}\sum_{m=0}^{M-1}\sum_{k=-K}^{K}\exp\left[j\pi \frac{\left(\cot\alpha U^{2}-2\csc\alpha Uk+\cot\alpha k^{2}\right)}{{\left(2\Delta x\right)}^{2}}-j2\pi \left(\omega_{m}\frac{k}{2}+\frac{\mu_{0}}{2}{\left(\frac{k}{2}\right)}^{2}\right)\right]\\ =\frac{B_{\alpha }}{2\Delta x}\exp\left[j\pi \cot\alpha {\left(\frac{U}{2\Delta x}\right)}^{2}\right]\sum_{m=0}^{M-1}\sum_{k=-K}^{K}\exp\left[j\pi \left(-\omega_{m}-\frac{2U\csc\alpha }{{\left(2\Delta x\right)}^{2}}\right)k+j\pi \left(-\frac{\mu_{0}}{4}+\frac{\cot\alpha }{{\left(2\Delta x\right)}^{2}}\right)k^{2}\right] \end{array}$$

From Equation (Equation 12), we can see that x can be reformulated into multiple (precisely say *M*) impulses only for a particular $\alpha_{0}$($\cot\alpha_{0}=-K\mu_{0}$) in the FrFT domain when K→∞. After peak searching, the peak coordinates $\left(\alpha_{B},U_{Bm}\right)$ in the FrFT domain can be utilized as an estimator for spatial frequency $v\left(\theta \right)$ and DOA θ as:(13)$$\left\{\begin{array}{c} \hat{v}=-\frac{1}{M-1}\sum_{m=2}^{M}\left(U_{Bm}-U_{Bm-1}\right)\frac{\csc\alpha_{B}}{2Kf_{\Delta }}\\ \hat{\theta }=\arcsin\left(\frac{c\hat{v}}{d_{R}}\right) \end{array}\right.$$

However, since the number of antennas *K* is a limited value, there always some residual terms between the quasi peaks $\left(\alpha_{B},U_{Bm}\right)$ and real peaks $\left(\alpha_{0},U_{m}\right)$. In this paper, we define these residual terms as $\phi_{0}$ and $\varepsilon_{m}$, where $\alpha_{0}=\alpha_{B}+\phi_{0}$ and $U_{m}=U_{Bm}+\varepsilon_{m}$. Concerning on the influence of these residual terms to the estimation precision of DOA, we propose an iterative high-accuracy method to solve this problem.

## 3. Proposed Method

### 3.1. Estimation of Spatial Chirp Rate

As the analytical formulation of $\left|X_{\alpha_{B}}\left(U_{Bm}\right)\right|$ involves Fresnel integral formula [25], it is difficult to directly construct the estimator for $\phi_{0}$. Thus, we consider utilizing the Fractional Autocorrelation (FA) spectrum of $x\left(t\right)$ to form this estimator, which is defined as [26]:(14)$$\begin{array}{c} \chi_{\alpha }\left(\tau \right)=\int x\left(t+\frac{\tau }{2}\sin\alpha \right)x^{*}\left(t-\frac{\tau }{2}\sin\alpha \right)e^{2j\pi t\tau \cos\alpha }dt\\ =\int \mathrm{rect}\left(\frac{t}{T_{K}}\right)e^{j2\pi t\tau \left(\mu_{0}\sin\alpha +\cos\alpha \right)}\sum_{m_{i}=0}^{M-1}\sum_{m_{j}=0}^{M-1}e^{-j\pi \tau v\left(\theta \right)\sin\alpha \left(m_{i}-m_{j}\right)f_{\Delta }}e^{-j2\pi v\left(\theta \right)\left(m_{i}-m_{j}\right)f_{\Delta }t}dt \end{array}$$
where $T_{K}\overset{\Delta }{\;=}KT_{s}$.

Afterwards, we can calculate the detection statistic [26] interpreted as:(15)$$L\left(\alpha \right)=\int_{-\infty }^{\infty }\left|\chi_{\alpha }\left(\tau \right)\right|d\tau$$

Substituting Equation (Equation 14) into Equation(Equation 15) yields
(16)$$L\left(\alpha \right)=\left|\Gamma \left(\alpha \right)\right|\int_{-\infty }^{\infty }\left|T_{K}\mathrm{Sinc}\left[2\pi T_{K}\left(\mu_{0}\sin\alpha +\cos\alpha \right)\tau \right]\right|d\tau$$
where
(17)$$\Gamma \left(\alpha \right)=\int_{-\infty }^{\infty }\int_{0}^{T_{K}}\sum_{m_{i}=0}^{M-1}\sum_{m_{j}=0}^{M-1}e^{-j\pi \tau vf_{\Delta }\sin\alpha \left(m_{i}-m_{j}\right)}e^{-j2\pi tvf_{\Delta }\left(m_{i}-m_{j}\right)}dtd\tau$$

We can ignore the $\Gamma \left(\alpha \right)$ in the following derivation as this term does not involve $\mu_{0}$. Therefore, we can estimate $\mu_{0}$ by locating the peak of $L\left(\alpha \right)$, namely:(18)$${\hat{\mu }}_{0}=-\cot\alpha {|}_{\alpha =\alpha_{0}}$$
where the coordination of the peak is given by $\alpha_{0}=\arg\max\left\{L\left(\alpha \right)\right\}$.

However, the estimation performance is affected by the grid size of searching, say Δα, as is demonstrated in Figure 2. To be specific, the actual residual term $\phi_{0}$ between the $\alpha_{0}$ and $\alpha_{B}$ is also defined by Δα, which is given by:(19)$$\alpha_{0}=\alpha_{B}+\phi_{0}=\alpha_{B}+\delta_{0}\Delta \alpha$$
where $\delta_{0}\in \left[-0.5,0.5\right]$. Therefore, the fine estimation is now equivalent to obtain an estimate of $\delta_{0}$. Plugging in Equations (Equation 18) and (Equation 19), after some trigonometric derivation, we can define the FA coefficient as:(20)$$L_{P}=L\left(\alpha_{B}+P\Delta \alpha \right)=\int_{-\infty }^{\infty }\left|T_{K}\mathrm{Sinc}\left[2\pi T_{K}\csc\alpha_{0}\sin\left(\left(P-\delta_{0}\right)\Delta a\right)\tau \right]\right|d\tau$$
where $L_{p}\left(P=\pm 0.5\right)$ calculates the interpolation coefficient at the both edges of $\alpha_{B}$. Afterwards, we introduce the error mapping formulation through Algorithm 1 of [27] (see Table I in [7] for more information), which is defined as:
(21)h1δ=ReL0.5+L−0.5L0.5−L−0.5≈12δ0

It is worth noting that Equation (Equation 21) needs a small enough Δα, then the following approximations can be utilized: $\sin\left(\delta \Delta \alpha \right)\approx \delta \Delta \alpha$ and $\sin\left[T_{K}\left(0.5-\delta \right)\Delta \alpha \pi \csc\tilde{\alpha }\tau \right]\approx \sin\left[T_{K}\left(0.5+\delta \right)\Delta \alpha \pi \csc\tilde{\alpha }\tau \right]$. Thus, we can construct the estimator ${\hat{\delta }}_{0}=\frac{1}{2h_{1}\left(\delta_{0}\right)}$ for $\delta_{0}$. Then, an iterative process can be combined to improve the estimation accuracy by updating $\alpha_{B}$ after each iteration, which will be shown in Section 3.3.

### 3.2. Estimation of Spatial Frequency

Firstly, following Equation (Equation 12), we consider one component, say *m*, of the OFDM-LFM signal with a well estimated spatial chirp rate $-\cot{\hat{\alpha }}_{0}\approx K\mu_{0}$. Thus, Equation (Equation 12) can be rewritten as:(22)$$X_{{\hat{\alpha }}_{0}}\left(\frac{U}{2\Delta x}\right)=\frac{B_{{\hat{\alpha }}_{0}}}{2\Delta x}e^{j\pi \cot{\hat{\alpha }}_{0}{\left(\frac{U}{2\Delta x}\right)}^{2}}\sum_{k=-K}^{K}e^{j\pi k\left(-\omega_{m}-\frac{2U\csc{\hat{\alpha }}_{0}}{{\left(2\Delta x\right)}^{2}}\right)}$$

As we analyze in Section 2, the coordination estimated from the discrete searching is bias from the actual value with the finite *K*. Hence, at the quasi peak $\left({\hat{\alpha }}_{0},U_{Bm}\right)$, $X_{{\hat{\alpha }}_{0}}\left(\frac{U_{Bm}}{2\Delta x}\right)$ equals:(23)$$X_{{\hat{\alpha }}_{0}}\left(\frac{U_{Bm}}{2\Delta x}\right)=\frac{B_{{\hat{\alpha }}_{0}}}{2\Delta x}e^{j\pi \cot{\hat{\alpha }}_{0}{\left(\frac{U_{Bm}}{2\Delta x}\right)}^{2}}\sum_{k=-K}^{K}e^{j\pi k\left(-\omega_{m}-\frac{U_{Bm}\csc{\hat{\alpha }}_{0}}{2K}\right)}$$

Substituting the real value $\omega_{m}=-\frac{\csc{\hat{\alpha }}_{0}}{2K}U_{m}$ and $U_{m}=U_{Bm}+\varepsilon_{m}$ into Equation (Equation 23), we can rewrite it as:(24)$$X_{{\hat{\alpha }}_{0}}\left(\frac{U_{Bm}}{2\Delta x}\right)=\frac{B_{{\hat{\alpha }}_{0}}}{2\Delta x}e^{j\pi \cot{\hat{\alpha }}_{0}{\left(\frac{U_{Bm}}{2\Delta x}\right)}^{2}}\sum_{k=-K}^{K}e^{j\pi k\frac{\varepsilon_{m}\csc{\hat{\alpha }}_{0}}{2K}}$$

Similar to the approach in Section 3.1, we can obtain $X_{{\hat{\alpha }}_{0}}\left(\frac{U_{Bm}\pm P}{2\Delta x}\right)$ as:(25)$$X_{{\hat{\alpha }}_{0}}\left(\frac{U_{Bm}\pm P}{2\Delta x}\right)=\Gamma^{\prime }\left({\hat{\alpha }}_{0},U_{Bm}\pm P\right)\left[\frac{e^{-j\pi \frac{\left(\varepsilon_{0}\mp P\right)\csc{\hat{\alpha }}_{0}}{2}}\left(1-e^{j\pi \left(\varepsilon_{0}\mp P\right)\csc{\hat{\alpha }}_{0}}\right)}{1-e^{j\pi \frac{\left(\varepsilon_{0}\mp P\right)\csc{\hat{\alpha }}_{0}}{2N}}}\right]$$
where $\Gamma^{\prime }\left({\hat{\alpha }}_{0},U_{Bm}\pm P\right)=\frac{B_{{\hat{\alpha }}_{0}}}{2\Delta x}e^{j\pi \cot{\hat{\alpha }}_{0}{\left(\frac{U_{Bm}\pm P}{2\Delta x}\right)}^{2}}$.

When $\left(\varepsilon_{m}\mp P\right)\ll N$, we can approximate Equation (Equation 25) by using the first order Taylor expansion at x=0 of $1-e^{x}\approx x$. Then, similarly to Section 3.1, we could also construct the error mapping through this approximation as:(26)$$X_{{\hat{\alpha }}_{0}}\left(\frac{U_{Bm}\pm P}{2\Delta x}\right)=\Gamma^{\prime }\left({\hat{\alpha }}_{0},U_{Bm}\pm P\right)\left[\frac{e^{-j\pi \frac{\left(\varepsilon_{0}\mp P\right)\csc{\hat{\alpha }}_{0}}{2}}\left(1-e^{j\pi \left(\varepsilon_{0}\mp P\right)\csc{\hat{\alpha }}_{0}}\right)}{1-e^{j\pi \frac{\left(\varepsilon_{0}\mp P\right)\csc{\hat{\alpha }}_{0}}{2N}}}\right]$$

Hence, we can similarly obtain an estimator ${\hat{\delta }}_{m}=0.5h_{2}\left(\delta_{m}\right)$ for the residual term ${\hat{\varepsilon }}_{m}$, and combine an iterative process to improve its accuracy.

### 3.3. Iterative DOA Estimation for OFDM-LFM

In this subsection, the estimators of spatial chirp rate and spatial frequency are combined to estimate the DOA for OFDM-LFM signals. Due to the fact that the FrFT is characterized by linear transformations [28], the major estimation bias between multi-component and mono-component signals through the FrFT based algorithm is caused by the energy leakage from the multi-component. To adapt the above process to the multi-component scenario, we introduce the CLEAN algorithm [27]. Firstly, the noise-free actual fractional coefficient ${\tilde{X}}_{{\hat{\alpha }}_{0},m}\left(\left({\hat{U}}_{m}\pm P\right)/2\Delta x\right)$ of the mth OFDM component is defined as: (27)$$\begin{array}{c} {\tilde{X}}_{{\hat{\alpha }}_{0},m}\left(\frac{{\hat{U}}_{m}\pm P}{2\Delta x}\right)=DFRFT_{\left({\hat{\alpha }}_{0},{\hat{U}}_{m}\pm P\right)}\left(x\left[k\right]\right)\\ =X_{{\hat{\alpha }}_{0},m}\left(\frac{{\hat{U}}_{m}\pm P}{2\Delta x}\right)+\sum_{l=1,\left(l\neq m\right)}^{M}{\overset{⌣}{\;X}}_{{\hat{\alpha }}_{0},l}\left(\frac{{\hat{U}}_{m}\pm P}{2\Delta x}\right) \end{array}$$
where $\begin{array}{c} {\overset{⌣}{\;X}}_{{\hat{\alpha }}_{0},l} \end{array}(({\hat{U}}_{m}\pm P)/2\Delta x)$ denote the energy leakage from the other *M* − 1 OFDM components, which can be calculated by: (28)$$\begin{array}{c} {\overset{⌣}{\;X}}_{{\hat{\alpha }}_{0},l} \end{array}\left(\frac{{\hat{U}}_{m}\pm P}{2\Delta x}\right)=A_{l}\mathrm{DFrF}T_{\left({\hat{\alpha }}_{0},{\hat{U}}_{m}\pm p\right)}\left({\hat{s}}_{l}\left[n\right]\right)=A_{l}\frac{B_{{\hat{\alpha }}_{0}}}{2\Delta x}e^{j\pi \cot{\hat{\alpha }}_{0}{\left(\frac{{\hat{U}}_{m}\pm P}{2\Delta x}\right)}^{2}}\sum_{n=-N}^{N}e^{j\pi n\left(\frac{\left({\hat{U}}_{l}-{\hat{U}}_{m}\mp P\right)\csc{\hat{\alpha }}_{0}}{2N}\right)}$$
where $A_{l}$ is the amplitude of the $l\mathrm{th}\left(l=1,\ldots ,M\right)$ component.

Then, the target fractional coefficient ${\hat{X}}_{{\hat{\alpha }}_{0},m}$ can be separated from the mixed term ${\tilde{X}}_{{\hat{\alpha }}_{0},m}$ by subtracting the leakages as: (29)$${\hat{X}}_{{\hat{\alpha }}_{0},m}\left(\frac{{\hat{U}}_{m}\pm P}{2\Delta x}\right)={\tilde{X}}_{{\hat{\alpha }}_{0},m}\left(\frac{{\hat{U}}_{m}\pm P}{2\Delta x}\right)-\sum_{l=1,\left(l\neq m\right)}^{M}{\overset{⌣}{\;X}}_{{\hat{\alpha }}_{0},l}\left(\frac{{\hat{U}}_{m}\pm P}{2\Delta x}\right)$$

According to the above derivation, an iteration-based method to accomplish the DOA estimation for OFDM-LFM signal is demonstrated in Algorithm 1.

**Algorithm 1:** Proposed FII-DFrFT DOA Estimation Method.

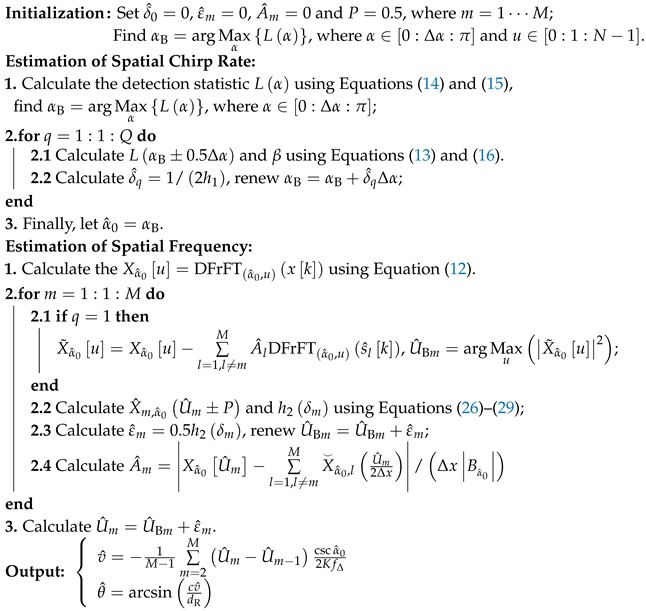



## 4. Performance Evaluation

In this section, we report our numerical evaluation through a Monte-Carlo simulation. Since the DOA estimation performance is mainly dependent on three factors, which are the signal-to-noise-ratio (SNR), the incidence angle (θ) and the component number (*M*), we evaluate the estimation performance with respect to these factors in a realistic case.

Consider a coherent MIMO radar (e.g. MIMO-SAR) which employs wideband OFDM-LFM signal. The simulation parameters of this MIMO radar and our SEDF system are listed in Table 1. It is worth noting that we assume the pulse width ($T_{P}$) of the OFDM-Signal is greater than $KT_{s}$, thus our method can function well. Moreover, we assume the far field sources whose initial phase is uniformly distributed within $\left[0,2\pi \right)$, and we take the thermal noise into consideration, which is modeled as zero-mean Gaussian with variance $\sigma_{n}^{2}=1$. Additionally, in all simulations, 1000 independent runs are conducted to calculate the Normalized Root Mean Square Error (NRMSE) and Root Mean Square Error (RMSE).

(a) *DOA Estimation versus SNR*


In this simulation, we evaluate the DOA estimation performance with respect to the SNR, while the DOA θ is set as 30 deg. For the sake of comparison, we also simulate the following approaches, Incoherent Signal-subspace Method Conventional Beam Forming (ISM-CBF) [29], Coherent Signal-subspace Method Linearly Constrained Minimum Variance (CSM-LCMV) [11], Rotational Signal Subspace Sparse Asymptotic Minimum Variance (RSS-SAMV) [12] and Sparse Iterative Covariance-based Estimation (SPICE) [9] As these existing approaches are designed for single wideband LFM signal, here, we consider the intercepted signal that received by our switched-element system a mono-component wideband LFM signal (M=1). Then, the above approaches and FII-FrFT are utilized to process the output signal and obtain the DOA estimation results, respectively. These results are collected and organized to NRMSE curves, which are shown in Figure 3. These curves reveal that our FII-FrFT method outperforms most mentioned approaches when SNR is beyond −8 dB. However, the NRMSE curve of FII-FrFT remains stable when SNR is beyond 12 dB and suffers a stable estimation bias, which is caused by the approximations that we employed in the theoretical derivations of Section 3.1 and the off-grid effect. On the other side, although the RSS-SAMV performs best in this simulation, its implementation will consume much more hardware resource (*K* receiving channels) and computational resource [12].

(b) *DOA Estimation versus Real Incident Angle and Component Number M*


In this simulation, we focus on the DOA estimation performance as the function of the real direction θ within $\left[10,70\right]$ degree by the FII-FrFT. We also consider the intercepted OFDM-LFM signals consist different component numbers $M=\left[2,3,4\right]$. For intuitional comparison with different OFDM-LFM signals, we define a different SNR in this subsection as $\rho =10\mathrm{lg}\left(MA^{2}/\sigma_{n}^{2}\right)$. The root mean square error (RMSE) of DOA estimation results at SNR = 10 dB are given in Figure 4. Firstly, we can see from Figure 4 that our proposed method can handle the OFDM-LFM radar signal well, while its component number affects the RMSE slightly. Secondly, the periodic variation of RMSE curves in Figure 4 reflects the off-grid effect in the fixed searching interval on estimation performance, which is in coincidence with our theoretical analysis in Section 3.2 and the simulation results in Reference [15]. This bias can be reduced by decreasing, i.e., using a denser grid, but it will also lead to the expensive price of computational load. Therefore, our DOA estimation method has to reach a compromise between accuracy and cost.

## 5. Conclusions

In this paper, a FII-DFrFT based SEDF system was introduced to improve the DOA estimation performance considering the wideband OFDM-LFM signals. The steering vector was reformulated followed by the iterative interpolation in both FA and DFrFT spectrum. Numerical simulations illustrated the validity and superiority of our algorithm compared with some other wideband DOA estimation approaches like ISM-CBF, CSM-LCMV, RSS-SAMV and SPICE. On the other hand, in the practice scenario, the modulated parameters of un-cooperative MIMO radar are generally unknown. This will cause the DOA estimation to be possibly ambiguous. Fortunately, taking advantage of a flexible switching interval, we can design a multi-interval SEDF system to resolve this ambiguity. Finally, the estimation bias caused by the off-grid effect and approximation are also of interest and will be the subject of our further investigation.

## Figures and Tables

**Figure 1 sensors-19-00132-f001:**
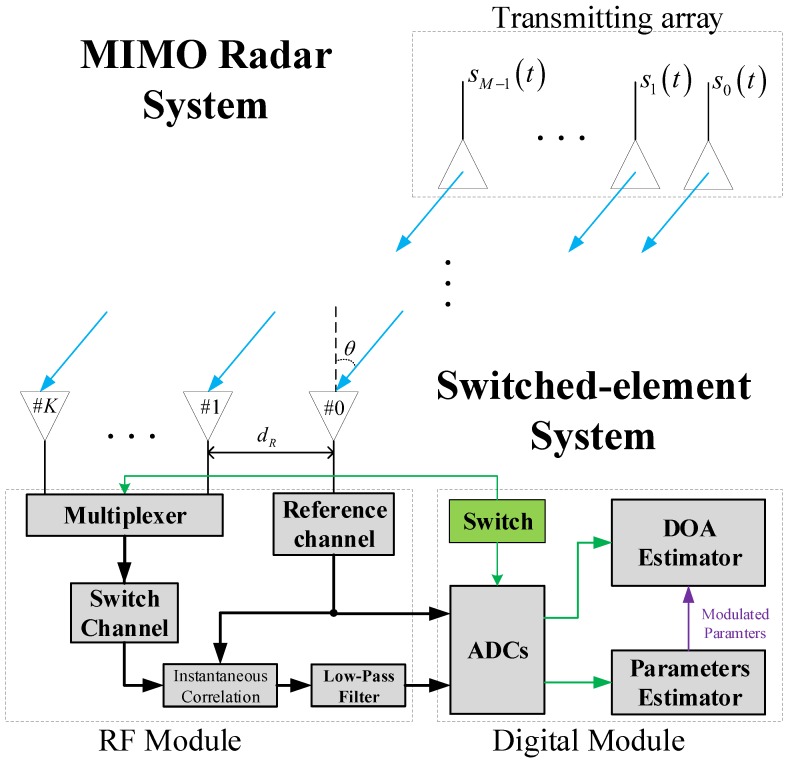
Block diagram of the Switched-Element Direction Finding System (RF is short for the Radio Frequency, ADCs is short for Analog to Digital converters) and Multiple-input multiple-output radar system.

**Figure 2 sensors-19-00132-f002:**
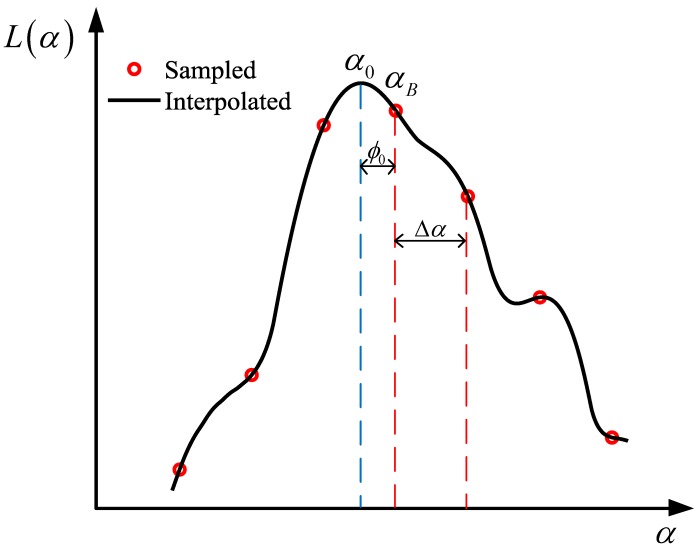
Demonstraction on the effect of the off-grid.

**Figure 3 sensors-19-00132-f003:**
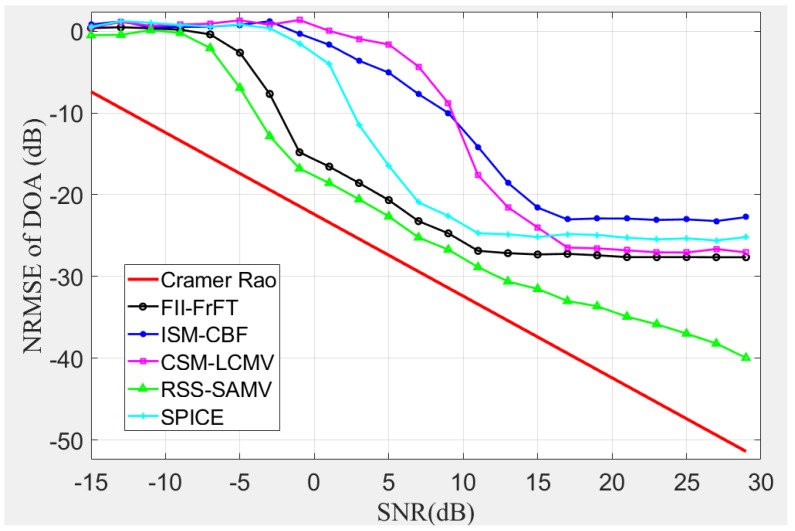
Normalized root mean square error (NRMSE) of DOA versus the signal-to-noise ratio (SNR). Cramer Rao, FII-FrFT, ISM-CBF, CSM-LCMV, RSS-SAMV and SPICE.

**Figure 4 sensors-19-00132-f004:**
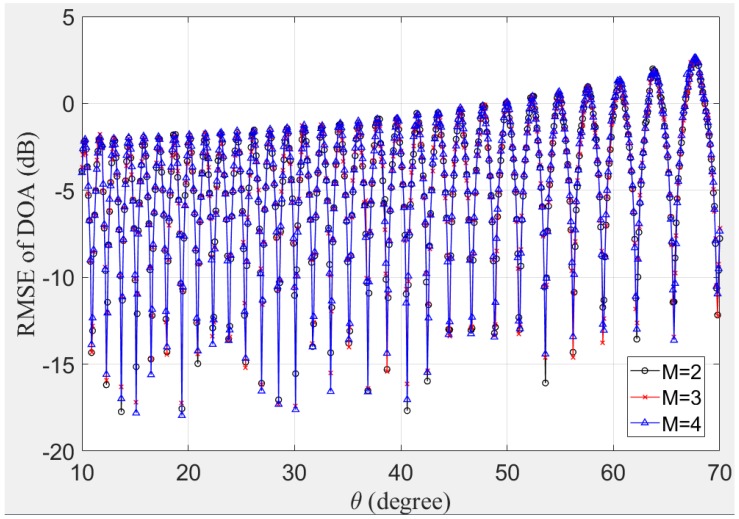
Root mean square error (RMSE) of DOA versus the incident angle.

**Table 1 sensors-19-00132-t001:** Parameter Settings.

**MIMO Radar Parameters**	Number of antennas *M*	1–4
	Pulse width $T_{p}$	20 μs
	Carrier frequency $f_{0}$	10 GHz
	Chirp rate $\gamma_{0}$	20 MHz/μs
	Frequency step $f_{\Delta }$	400 MHz
**Switched-element System Parameters**	Number of ULA *K*	128
	Carrier frequency $f_{0}$	10 GHz
	Interspace of ULA $d_{R}$	0.015 m
	Switching interval $T_{s}$	0.1 μs
	Searching interval Δα	0.01
	Iteration number *Q*	3

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
