# Peer review of "A Switched-Element System Based Direction of Arrival (DOA) Estimation Method for Un-Cooperative Wideband Orthogonal Frequency Division Multi Linear Frequency Modulation (OFDM-LFM) Radar Signals"

_sensors, 2019, doi:10.3390/s19010132_

Reviewer 1 Report

A very interesting paper. However comparisons with existing approaches should be more detailed. Only NRMSE has been compared. What about average over changing target scenario?

Please consider the broadband beamforming approaches in van Trees book. Include a discussion on those approaches in the Introduction.

The switched element system could be thought of as a random array. How would this compare with results of e.g. "High spatial resolution radar using thinned arrays", IEEE Radar Conf 2017?

Author Response

Please see the "Reply_to_Reviewer2.pdf"

Reviewer 2 Report

Dear authors,

thank you for your interesting work.

Some small comments.

With fig 1. it would be nice to sketch also the MIMO Radar to intercept. You could e.g. mark antenna m from 0 to M.

Line 48 ff

"shown in Figure 1." - please remove dot after Figure or write Fig.

Page 3 top

here you are using the term "SOI" the first time, but you did not introduce this abbreviation.

A general comment. It is somehow confusing that in some calculations you refer to RF and in some to baseband. I understand that in wideband signals, the benefit from baseband representation is less. However I would stick to one view.

You could place the mid of baseband at f0, so that all LFM ramps are one-sided, so analytically…

That does not pose any restrictions.

page 3, text between eq 5 and 6

You are refeing to a lowpass filter to let low frequency components pass.

Looking at the phase terms, we see that Phi_m is related to fm, thus fo, so very high.

On the opposite the term j 2 pi (m-m') is related to fdelta, so something much smaller in frequency. This let me to the conclusion you wanted to pass the high frequency components to get rid of low frequency cross terms.

Also in your study Table 1 you Mention f0 as 10 GHz and fdelta as 400 MHz.

Please clarify this inconsistency.

page 8, line 76

"is mainly dependent on three factors"

Regards

reviewer

Author Response

Reply_to_Reviewer2.pdf.

Reviewer 3 Report

Interesting and well written paper.

It would be interesting to show the results for more then one target in terms of the beampattern.

Author Response

After carefully checking the whole manuscript, we revised numbers of grammatically wrong sentences. Moreover, we added and deleted some words so that the manuscript maybe “read” better. We also adjusted the structure of some sentences, which make them more logical to read. It is worth noting that we are open to pay for professional English editing service so that we can guarantee the quality of this manuscript. We are willing to have further discussion on this manuscript with the reviewer and editor to make it more suitable for publishing.

Point 1: It would be interesting to show the results for more than one target in terms of the beampattern.

Response 1: Thanks for the honorable Reviewer’s suggestion. We all authors totally agree with it and we will extend the research of the SEDF system to adapt it for multiple targets environment in our future work.
